# Carvacrol Attenuates Hippocampal Neuronal Death after Global Cerebral Ischemia via Inhibition of Transient Receptor Potential Melastatin 7

**DOI:** 10.3390/cells7120231

**Published:** 2018-11-26

**Authors:** Dae Ki Hong, Bo Young Choi, A Ra Kho, Song Hee Lee, Jeong Hyun Jeong, Beom Seok Kang, Dong Hyeon Kang, Kyoung-Ha Park, Sang Won Suh

**Affiliations:** 1Department of Physiology, College of Medicine, Hallym University, Chuncheon 24252, Korea; zxnm01220@gmail.com (D.K.H.); bychoi@hallym.ac.kr (B.Y.C.); rnlduadkfk136@hallym.ac.kr (A.R.K.); sshlee@hallym.ac.kr (S.H.L.); jd1422@hanmail.net (J.H.J.); ttiger1993@gmail.com (B.S.K.); ehdgus2728@naver.com (D.H.K); 2Division of Cardiovascular Diseases, Hallym University Sacred Heart Hospital, Anyang 14068, Korea; pkhmd@naver.com

**Keywords:** global cerebral ischemia, carvacrol, transient receptor potential melastatin 7, zinc, neuronal death

## Abstract

Over the last two decades, evidence supporting the concept of zinc-induced neuronal death has been introduced, and several intervention strategies have been investigated. Vesicular zinc is released into the synaptic cleft, where it then translocates to the cytoplasm, which leads to the production of reactive oxygen species and neurodegeneration. Carvacrol inhibits transient receptor potential melastatin 7 (TRPM7), which regulates the homeostasis of extracellular metal ions, such as calcium and zinc. In the present study, we test whether carvacrol displays any neuroprotective effects after global cerebral ischemia (GCI), via a blockade of zinc influx. To test our hypothesis, we used eight-week-old male Sprague–Dawley rats, and a GCI model was induced by bilateral common carotid artery occlusion (CCAO), accompanied by blood withdrawal from the femoral artery. Ischemic duration was defined as a seven-minute electroencephalographic (EEG) isoelectric period. Carvacrol (50 mg/kg) was injected into the intraperitoneal space once per day for three days after the onset of GCI. The present study found that administration of carvacrol significantly decreased the number of degenerating neurons, microglial activation, oxidative damage, and zinc translocation after GCI, via downregulation of TRPM7 channels. These findings suggest that carvacrol, a TRPM7 inhibitor, may have therapeutic potential after GCI by reducing intracellular zinc translocation.

## 1. Introduction

Transient cerebral ischemia results from decreased blood flow into the brain, primarily through blood vessels, such as the bilateral common carotid artery and vertebral artery. Due to this interruption in blood flow, complex cellular mechanisms of degeneration ensue if there is no intervention [1]. Lack of blood supply can lead to oxygen deprivation and insufficient nutrient supply [2]. Ischemic brain injury can be divided into two main categories. Focal cerebral ischemia occurs when a blood clot and thrombus forms an occlusion in the blood vessels. Focal cerebral ischemia entails a focused reduction of cerebral blood flow to specific brain regions [3]. Unlike focal cerebral ischemia, global cerebral ischemia entails a reduction in blood flow toward the entire brain, due to cardiac infarction. This brain insult occurs generally in patients who have a variety of clinical symptoms, and a generally degraded physiological condition involving asphyxia, cardiac arrest, and shock [4,5,6].

During ischemia, diminished blood flow toward the brain disrupts neuronal energy metabolism, and disturbs intra- and extracellular ionic homeostasis [7]. Subsequently, the ischemic condition leads to lipid peroxidation [8], microglial activation [9], zinc accumulation [10], neuronal degeneration, and other complex cellular processes [11,12]. Although the reduced cerebral blood flow can be recovered by reperfusion, this event can separately induce severe secondary injuries, such as increased reactive oxygen species (ROS) production [4]. The accumulation of excessive metal ions, such as calcium and zinc, into the post-synaptic intracellular space from the pre-synaptic vesicle has been suggested as causal in events upstream of ischemic cascades. Excessive calcium influx can cause neuronal dysfunction, which can ultimately result in neuronal cell death [13].

Zinc is an abundant metal ion in the brain, and it regulates several physiological functions, such as cell proliferation and differentiation, protein synthesis, and cellular signal recognition, as well as being involved in many kinds of enzymatic cascades [14]. In addition, zinc is loosely bound to proteins, where it can act as a component of the catalytic site and modulate morphological and structural capacity in the enzyme [15]. Together, this circumstantial evidence suggests that zinc plays a key role in the homeostatic maintenance of the central nervous system. The pool of chelatable zinc is located in the pre-synaptic vesicles, and is released into the synaptic cleft during neuronal activation (e.g., action potential generation) [16]. Under normal conditions, released chelatable zinc can undergo reuptake into pre-synaptic neurons, in order to maintain a consistent intra- or extra-neuronal zinc balance. However, in the case of neurological injury, toxic levels of chelatable zinc can be released from the synaptic vesicle, and can then dissociate from intracellular proteins. This phenomenon contributes to neuronal death and causes zinc-induced neurotoxicity [17,18]. In previous studies, the potential of exogenous zinc to promote neurotoxic outcomes was confirmed in a variety of neurological disorders [19,20]. Accordingly, previous studies by our group have demonstrated that toxic concentrations of intraneuronal zinc accumulation can lead to neuronal death after cerebral ischemia [21], epilepsy [22], and hypoglycemia [23].

Transient receptor potential melastatin 7 (TRPM7) channels are metal ion-permeable, non-selective cation channels that are expressed in almost all tissues [24], and which form a subunit of the large TRP channel [25]. Activated TRPM7 channels have a role in both physiological and pathophysiological functions that regulate divalent cation (Mg^2+^, Ca^2+^, Zn^2+)^ homeostasis [17,26]. Under the neurological injury setting, extracellular Ca^2+^ influx into the intracellular space through TRPM7 produces reactive oxygen species, resulting in feed-forward activation of TRPM7 that leads to neurodegeneration [27]. TRPM7 has several electrophysiological characteristics that might be clues for identifying a solution to damage arising from cerebral ischemia [1].

A role for TRPM7 in ischemia-induced neuronal damage has been previously reported [28,29]. In a past in vitro study, under oxygen-glucose deprived conditions, an increase in Ca^2+^ entry that led to neuronal cell death was observed. In another preceding study, the use of glutamate receptor inhibitors and voltage-gated Ca^2+^ channel blockers as neuroprotective agents was established [27]. By using TRPM7 shRNA viral vectors [29], it was demonstrated that ischemic cell death mechanisms were also regulated by TRPM7. These findings suggested that, alongside voltage-gated calcium channels and the glutamate receptor-like proton-sensitive channel, the TRPM7 channel has a critical role in mediating ischemic brain injury. Interestingly, TRPM7 channels have recently been reported to regulate Zn^2+^ toxicity in cultured hippocampal neurons [17]. Additionally, the same study showed that Zn^2+^ displays neurotoxicity, which was decreased by the application of TRPM7 blockers [Gd^3+^ and 2-Aminoethoxydiphenyl borate (2-APB)].

Carvacrol is a monoterpene that has been associated with neuroprotection in several neurological disorders, involving traumatic brain injury [30], epilepsy [31], and hemiparkinsonism [32]. Furthermore, this naturally-occurring agent has anti-inflammatory effects in interleukin-10 knockout mice, which indicates the involvement of inflammatory mechanisms [33]. In a previous study, Pamas and colleagues demonstrated that carvacrol blocks the over-expression of the TRPM7 channels at the hippocampal CA1–CA3 regions in mammalian cells [34].

Following the logic that carvacrol has neuroprotective effects against various kinds of neurological damage, we hypothesized that carvacrol can attenuate ischemia-induced neuronal death by reducing zinc-induced neurotoxicity. Here, we demonstrate that intraperitoneal administration of carvacrol is associated with a significant reduction in hippocampal damage after global cerebral ischemia. Therefore, we suggest that carvacrol may be an ideal therapeutic tool for preventing global cerebral ischemia-induced neuronal death.

## 2. Results

### 2.1. Carvacrol Attenuates Global Cerebral Ischemia-Induced Hippocampal Neuron Death

To comprehensively evaluate whether carvacrol promotes neuroprotection after global cerebral ischemia-induced hippocampal neuronal death, experimental animals were given daily intraperitoneal injections of carvacrol (50 mg/kg) or 0.1% dimethyl sulfoxide (DMSO; diluted with normal saline) for three days after global cerebral ischemia (GCI). At three days after GCI, histological evaluation to detect degenerating neurons was performed in the hippocampal subiculum (Subi), Cornus Ammonis 1 (CA1), and Cornus Ammonis 2 (CA2) regions. The results showed that Fluoro-Jade B (FJB)-stained hippocampal neurons emerged in the Subi, CA1, and CA2 regions (Figure 1A). Accordingly, there was no fluorescence signal detected in the sham-operated hippocampal slices. The number of degenerating neurons in the GCI-exposed groups was dramatically increased compared to the sham-operated groups. When compared with the vehicle treated groups, groups given carvacrol showed dramatically reduced hippocampal neuronal death. Figure 1B represents the quantified FJB (+) neurons in the Subi, CA1, and CA2 regions. Administration of carvacrol showed an approximate 54% reduction in the number of FJB (+) neurons in the Subi (GCI vehicle, 191.1 ± 8.7; GCI carvacrol, 88.4 ± 15.4), 67% in the CA1 (GCI vehicle, 159.4 ± 21.6; GCI carvacrol, 53.7 ± 20.2), and 65% in the CA2 (GCI vehicle, 181.2 ± 6.3; GCI carvacrol, 65.1 ± 54.3) region, when compared to the vehicle group.

### 2.2. Carvacrol Administration Decreases Zinc Translocation to the Hippocampal Pyramidal Layer after Global Cerebral Ischemia

GCI induces extracellular Zn^2+^ accumulation from synaptic vesicles [35]. Released Zn^2+^ contributes to the exacerbation of ischemic brain damage by gaining access to the intracellular space through proton-sensitive cation channels, such as TRPM7 [1,36], indicating that zinc homeostasis can be disturbed by neurological disorders. Therefore, intracellular zinc concentration is tightly controlled by numerous types of proteins, regulated zinc transporters, and zinc sensors [37]. To determine whether carvacrol administration regulates zinc concentration at the extra- and intra-cellular space after GCI, we conducted TSQ (6-methoxy-8-*p*-toluenesulfonamido-quinoline) staining, as this is a specific indicator for intracellular free zinc levels. Compared with the sham-operated groups, a sizable number of TSQ fluorescence (+) neurons were seen in the hippocampal pyramidal layer after GCI. Interestingly, after GCI, the group that was given carvacrol had reduced Zn^2+^ translocation in the hippocampal pyramidal layer (Figure 1C). Figure 1D represents the quantified TSQ fluorescence signal in CA1. Administration of carvacrol showed an approximate 36% reduction in Zn^2+^ translocation to the CA1 region (GCI vehicle, 23.96 ± 1.26; GCI carvacrol, 15.43 ± 1.66) when compared to the vehicle group.

### 2.3. Carvacrol Administration Reduced Global Cerebral Ischemia-Induced Hippocampal Oxidative Damage

Following GCI, oxidative damage occurred in several hippocampal regions, including the Subi, CA1, CA2, and others, by enhancing the production of reactive oxygen species (ROS). ROS can activate several cell death mechanisms and damage the plasma membrane [38]. In order to estimate the degree of ROS production and lipid peroxidation after GCI, we performed 4-hydroxy-2-nonenal (4HNE) staining to serve as a marker for oxidative damage [39]. In Figure 2A–C, the sham-operated groups that had 0.1% DMSO (diluted with normal saline) and carvacrol administrated showed no differences in oxidative damage between them, while a strong 4HNE fluorescence signal from each hippocampal region was observed in the GCI group. In contrast, carvacrol administration after GCI showed a significant reduction of the 4HNE fluorescence signal. Figure 2D shows the quantified 4HNE fluorescence signal in the Subi, CA1, and CA2. Administration of carvacrol gave an approximate 25% reduction in oxidative damage in the Subi (GCI vehicle, 18.17 ± 0.58; GCI carvacrol, 13.67 ± 0.19), 28% in the CA1 (GCI vehicle, 18.15 ± 1.46; GCI carvacrol, 13.14 ± 0.6), and 44% in the CA2 (GCI vehicle, 19.4 ± 0.42; GCI carvacrol, 10.88 ± 0.36), when compared to the vehicle group.

### 2.4. Carvacrol Administration Decreases Global Cerebral Ischemia-Induced Microglial Activation

GCI induces several pathophysiological changes, including an inflammatory and immune response, with microglia becoming activated and assuming an amoeboid morphology. Under normal conditions, resting state microglia maintain a regular distance from one another, and are evenly distributed throughout the central nervous system [40]. Cerebral ischemic insult leads to an enhanced neuroinflammatory response. Consequently, this leads to microglial proliferation and activation. To estimate the fluorescence signal corresponding to microglia activation after GCI, we performed ionized calcium-binding adaptor molecule 1 (Iba1) staining as a specific marker for microglia. In the sham-operated animals, vehicle and carvacrol treatment groups both displayed a similar number of Iba-1 positive microglial cells. However, activated microglia were detected in the hippocampal pyramidal layer after GCI. The increased number of microglial cells expressed after GCI was reduced by carvacrol administration (Figure 3A). Figure 3B shows the quantified Iba-1 positive microglial cells/mm^2^ in the CA1 region. Administration of carvacrol showed an approximate 54% reduction in the number of microglial cells in the CA1 (GCI vehicle, 312.73 ± 26.74; GCI carvacrol, 187.20 ± 14.12) when compared to the vehicle group.

### 2.5. Carvacrol Administration Inhibits Expression of Transient Receptor Potential Melastatin 7 Channels after Global Cerebral Ischemia

Transient receptor potential melastatin 7 channels have been shown to be widely expressed in mammalian cells and tissues [24]. Expression of these channels has been associated with several neurological diseases, such as cerebral hypoxia and ischemia [41]. In addition, various extracellular ions enter the intracellular space through this channel [1]. To test whether expression of TRPM7 channels was inhibited by using carvacrol, we conducted TRPM7 and NeuN staining. In Figure 4A, the GCI-operated groups were elevated for the TRPM7-associated fluorescence signal, and found to be greater than that of the sham-operated groups. This indicates that over-activated TRPM7 channels can lead to cerebral degeneration via an influx of metal ions. However, the TRPM7 inhibitor carvacrol significantly reduces TRPM7 expression after GCI. Figure 4B represents the quantified TRPM7 fluorescence signal in the hippocampal CA1 region. Consequently, administration of carvacrol showed an approximate 29% reduction of TRPM7 expression in the CA1 region (sham vehicle, 6.21 ± 0.37; sham carvacrol, 5.89 ± 0.67; GCI vehicle, 15.19 ± 0.62; GCI carvacrol, 10.92 ± 0.96).

## 3. Discussion

The present study investigated whether carvacrol administration had neuroprotective effects on global cerebral ischemia (GCI)-induced hippocampal neuron damage via the inhibition of transient receptor potential melastatin 7 (TRPM7) channels. Interestingly, carvacrol displays hippocampal neuroprotection after GCI, and significantly decreased GCI-induced cell death cascades, such as the degeneration of hippocampal neurons, oxidative damage via lipid peroxidation, and inflammatory mechanisms mediated by activation of microglia. Consequently, this research suggests that carvacrol administration shows potential as a therapeutic agent against GCI.

Under normal physiological conditions, homeostatic regulation of ionic gradients is well maintained and preserved throughout the nervous system. However, cerebral ischemic damage causes disruption of ionic homeostasis [7]. Ion entry through proton-sensitive cation channels, which includes voltage-sensitive Ca^2+^ channels, *N*-methyl-d-aspartate (NMDA) receptors [42], transient receptor potential (TRP) channels, and acid-sensing ion channels (ASIC), contributes to ionic misbalance. Several studies have demonstrated that ion channels are closely involved with the pathological mechanisms of cerebral ischemia and excitotoxicity [1,43]. One ischemic cascade involves zinc ion accumulation in hippocampal neurons, and contributes to zinc-induced neurotoxicity [10]. Understanding zinc toxicity is one of the key clues that may help solve the problem of cerebral injury.

Our previous studies have demonstrated that diverse brain injuries, such as global ischemia, traumatic brain injury, and hypoglycemia, all induce excessive zinc accumulation in hippocampal neurons [10,21,44,45]. In this setting, free zinc ions can translocate to post-synaptic neurons via proton-sensitive cation channels and transporters. TRPM7 is a class of cation-permeable ion channels that regulates biological processes, such as metal cation (Mg^2+^, Zn^2+^, Ca^2+^) homeostasis and neurotransmitter release [1]. Here, we focused on the finding that the TRPM7 channel is involved in the regulation of zinc ions. Hence, we hypothesized that inhibition of the TRPM7 channel would decrease zinc accumulation and prevent zinc-induced neuronal death. The present study was conducted to determine whether carvacrol, a TRPM7 inhibitor, has neuroprotective effects with regard to global cerebral ischemia-induced hippocampal neuronal death.

Carvacrol is a naturally occurring oil that is known to be an inhibitor of TRPM7 channels [34] and to protect against brain damage from hypoxic-ischemic injury [46]. These observations suggest that using carvacrol to inhibit TRPM7 channels and reduce zinc entry into the intracellular space may be a strategy to reduce brain damage after ischemia.

In addition, other TRP channels, such as TRPC (canonical), TRPV (vanilloid), and TRPA (ankyrin) have been associated with the regulation of membrane potential and intracellular Ca^2+^ concentration. These channels were activated by several changes in the surrounding environment, cellular stress, and receptor stimuli [25,47,48]. A previous study demonstrated that the TRPM2 channel was activated by ROS. Activated TRPM2 channels lead to extracellular Ca^2+^ influx, activate the PARP-1/PARG signaling pathway, and regulate caspase signaling [48,49]. Additionally, TRPC5 is directly activated by hydrogen peroxide and nitric oxide (NO) donors. Both redox conditions and drug administration affect TRPC5 function. Moreover, oxidative stress leads to activation and sensitization of the non-selective cation channel TRPV1 [47]. Taken together, many members of the TRP channel family are modulated by ROS, which may arise due to closely-related structural and functional similarities. However, carvacrol regulates some of these channels, such as TRPA1, TRPV3, and TRPM7 [50,51,52]. Thus, we suggested that TRP channels, particularly TRPM7, may be a potent target for preventing ROS-induced cellular damage.

To verify whether carvacrol administration after GCI has a neuroprotective effect, we performed histological evaluation by detecting degenerating hippocampal neurons using Fluoro-Jade B (FJB) staining. The number of FJB (+) hippocampal neurons from the subiculum, CA1, and CA2 were significantly reduced by carvacrol administration. Neuronal free zinc levels were detected by TSQ (*N*-(6-methoxy-8-quinolyl)-*para*-toluenesulfonamide) staining. Since several studies have shown that inhibition of TRPM7 channels reduces extracellular zinc entry into hippocampal neurons, we evaluated brain sections after GCI. In the present study, we found that the number of TSQ (+) neurons in the hippocampus was dramatically decreased by carvacrol administration after GCI. This result indicates that intraneuronal free zinc accumulation was decreased by inhibiting TRPM7 channels using carvacrol, which led to reduced zinc-induced neurotoxicity. Ischemic damage induces dysfunction of the mitochondria, and leads to ROS production and lipid peroxidation [53]. ROS has been determined to have a potentially fatal impact on the central nervous system, and 4HNE (4-hydroxy-2-nonenal) staining was used to detect lipid peroxidation after GCI. In the present study, we found that carvacrol administration reduced hippocampal oxidative damage after GCI. GCI leads to the activation of microglia in the hippocampal regions. To test whether microglial activation is prevented by carvacrol, ionized calcium-binding adaptor molecule (Iba1) staining was implemented in the studied brain sections. Activated microglia staining was significantly increased in the GCI vehicle group. However, carvacrol administration significantly decreased microglial activation after GCI. Finally, we measured TRPM7 channel expression after GCI, and whether carvacrol could inhibit TRPM7 channel expression after ischemia. TRPM7 immunofluorescence staining was conducted for each experimental group. In the present study, we found that TRPM7 channel expression was increased after GCI. We also found that more free zinc influx from the extracellular space to the intraneuronal space occurred [36].

Therefore, we hypothesized that free zinc entering through TRPM7 channels into the intraneuronal space contributes to neurotoxicity in this setting. The result that activation of TRPM7 channel occurs in the context of zinc-induced cell damage has been reported previously [17]. We also found that carvacrol reduced various aspects of GCI-induced brain injury, such as neuronal degeneration, lipid peroxidation, microglial activation, and zinc accumulation. A proposed schematic diagram in Figure 5 illustrates our hypothesis that reducing zinc accumulation by the use of a specific TRPM7 channel inhibitor reduces GCI-induced hippocampal neuronal death.

Altogether, the present study demonstrates that carvacrol administration provides neuroprotection by decreasing zinc accumulation in neurons through the inhibition of the TRPM7 channel. Therefore, carvacrol may be a useful therapeutic agent for preventing stroke-induced neuronal death, and the TRPM7 channel is a novel target for treating neurological disorders.

## 4. Materials and Methods

### 4.1. Ethics Statement

This present research procedure was in accordance with the guidelines put forth by the Laboratory of Animal Care and Use Institute of Hallym University (Protocol # Hallym 2018-34). To minimize experimental-animal suffering, we used isoflurane anesthesia during all the research procedures.

### 4.2. Experimental Animals

In the present study, eight-week-old Sprague–Dawley male rats were used (280~320 g; DBL Co., Chungcheongbuk-do, Eumseong-gun, Korea). The animals were housed under a consistently maintained temperature (22 ± 2 °C), humidity (55 ± 5%), and an automatically controlled 12-hour light and dark cycle (lights turned on and off) with food (Purina, Gyeonggi, Korea), and water adlib.

### 4.3. Global Cerebral Ischemia Surgery

Commonly used global cerebral ischemia induction by bilateral common carotid artery occlusion (CCAO) is well-known and has been reported in a previous study [54]. The experimental rats received the global cerebral ischemic surgery under consistently maintained anesthetization using 2~3% isoflurane, were ventilated with 30% oxygen and 70% nitrous oxide, and were kept at a body temperature of 37 ± 1 °C using a heating pad [55]. A catheter filled with heparin was inserted into the femoral artery. Cannulation was used for the observation of arterial blood pressure and draining blood. We then revealed and transiently occluded the bilateral common carotid arteries that are located beside the tracheal muscle. In this situation, we used micro dissecting forceps and a surgical microscope (SZ61, Olympus, Shinjuku, Japan) to avoid stimulation of the vagus nerve and improve surgical accuracy. Electroencephalographic (EEG) probes were placed in bilateral burr holes. Systemic mean arterial pressure (MAP) was decreased within the range of 40 ± 10 mmHg by draining blood (7~10 cc) from the inserted catheter filled with heparin through the femoral artery. The exposed bilateral common carotid arteries were clamped with medical clamp (Serrefine, Fine Science Tolls, Foster City, CA, USA) when they were within the 40 ± 10 mmHg MAP range. The MAP and EEG were taken at 7 min after global cerebral ischemia induction, following the onset of isoelectricity [55]. 7 min later, blood circulation was reinstated by unclamping the occlusion, after which vitals, MAP, and EEG signals were monitored until they returned to baseline. Carvacrol (50 mg/kg) in 1.5~2 mL of 0.1% dimethyl sulfoxide (DMSO) was immediately administrated to the intraperitoneal space in the carvacrol treated groups. Control rats for these studies received an equal volume of vehicle (0.1% DMSO diluted with normal saline) alone. After the termination of the ischemia surgery, the animals were monitored for a few hours in a temperature-controlled incubator and provided with food and water ad libitum.

### 4.4. Carvacrol Administration

To test whether carvacrol has neuroprotective effects in global cerebral ischemia-induced hippocampal degeneration, the experimental animal groups were divided into the following: sham (vehicle, carvacrol) and global cerebral ischemia (vehicle, carvacrol). Carvacrol (50 mg/kg) was dissolved in 0.1% DMSO and administrated once per day for three days. The vehicle administrated groups had the same schedule with 0.1% DMSO (diluted with normal saline). All experimental animals were anesthetized and sacrificed at three days following global cerebral ischemia.

### 4.5. Brain Sample Preparation

To analyze the neuroprotection of carvacrol administration after GCI-induced brain damage, all experimental animals were sacrificed at three days after GCI. They were anesthetized with isoflurane (1.5~2%) following urethane (1.5 g/kg, IP). The deeply anesthetized experimental animals were transcardially perfused with 0.9% normal saline, followed by 4% paraformaldehyde for sample fixation. After the termination of perfusion, the whole brain was obtained and then immersed in 4% paraformaldehyde for one hour. For post-fixation, the brain sample was transferred to 30% sucrose solution for a few days. After the brain sample sank to the bottom of tube, 30 μm thickness sections were sliced using a cryostat for immunohistochemistry and immunofluorescence.

### 4.6. Evaluation of Hippocampal Degenerating Neurons

To determine whether carvacrol has neuroprotective effects against global cerebral ischemia-induced hippocampal neuronal death, we conducted the Fluoro-Jade B (FJB) staining procedure. Brain slices from all experimental groups were mounted on gelatin-coated slides (Fisher Scientific, Pittsburgh, PA, United States). These samples were then reacted with 100% ethanol, followed by 70%, and then distilled water for hydration. After hydration in distilled water, the samples were reacted with 0.06% potassium permanganate solution for 15 min, washed with distilled water once one minute, and then soaked in 0.001% FJB solution (Histo-chem Inc., Jefferson, AR, USA) for 30 min. Lastly, the samples were washed with distilled water for 1 min, three times. After washing, the slides were dried gently using an air flow machine (Daihan labtech, Gyeonggi, Korea) for approximately 3 or 4 h. The FJB-reacted slides were dehydrated in xylene and mounted with mounting media DPX (Sigma-Aldrich Co., St. Louis, MO, United States). The slides were observed using a microscope (blue fluorescence wavelength: 450–490 nm, Olympus, Shinjuku, Japan). To accurately count the hippocampal cells that had an FJB fluorescence signal, a blinded experimenter was needed. This experimenter counted hippocampal FJB cells in the bilateral subiculum, CA1, and CA2 areas. The pooled number of FJB positive cells was used for statistical analysis.

### 4.7. Evaluation of Hippocampal Zinc Translocation

To examine whether carvacrol administration reduces vesicular and intraneuronal free zinc translocation, 4% paraformaldehyde fixed fresh frozen brain slices were reacted with *N*-(6-methoxy-8-quinolyl)-*para*-toluenesulfonamide (TSQ; Molecular Probes, Eugene, OR, United States) [56]. The sham and GCI group rats were anesthetized deeply with 4~5% isoflurane, and the entire brain sample was obtained without transcardial perfusion. The obtained fresh whole brain was put on dry-ice powder for rapid freezing, and stored at –80 °C in the freezer. The frozen brain was coronally sliced at a 10 μm thickness, mounted on gelatin-coated slides, and then reacted with a TSQ solution containing 4.5 μM TSQ, 140 mM sodium barbital, and 140 mM sodium acetate [57].

### 4.8. Evaluation of Hippocampal Lipid Peroxidation

One cerebral ischemic cascade is reactive oxygen species (ROS) production via lipid peroxidation. To verify whether carvacrol administration reduces hippocampal lipid peroxidation after GCI, we performed 4HNE (4-hydroxy-2-nonenal) staining. An approximate protocol of 4HNE staining has been described in our previous studies [21,57]. Briefly, hippocampal (5~6) slices were reacted with 4HNE antiserum (diluted 1:500; Alpha Diagnostic Intl. Inc., San Antonio, TX, United States) in PBS involving 0.3% Triton X-100 for 15~18 h in a 4 °C maintained incubator. After reaction with the 4HNE antiserum they were rinsed with PBS for 10 min, three times. Then, the brain slices were incubated in Alexa Fluor 594 goat anti-rabbit IgG antiserum (diluted 1:250, Invitrogen, Carlsbad, CA, USA) and counter stained with DAPI (diluted 1:1000) for 2 h in succession. These samples were photographed using a fluorescence microscope (red fluorescence wave length: 590–617 nm). The 4HNE fluorescence signal was counted by using ImageJ software (NIH, Bethesda, MA, USA). The pooled 4HNE intensity was used in this study for statistical analysis in all groups.

### 4.9. Evaluation of Hippocampal Microglial Activation

Inflammatory responses are initiated and worsened by external damage such as cerebral ischemia [33]. To determine whether carvacrol can reduce hippocampal microglial activation after GCI, we performed staining against Iba1, a specific marker for microglia detection. The general staining protocol is as follows. Brain sections were incubated in Iba1 primary antibody (diluted 1:500, Abcam, Cambridge, UK) containing 0.3% Triton X-100 for 15~18 hours in a 4 °C maintained incubator. Next, they were reacted with Alexa Fluor 488-conjugated donkey, anti-goat, IgG secondary antibody (diluted 1:250; Invitrogen, Grand Island, NY, United States), and counter-stained with DAPI (diluted 1:1000) for two hours. The sections were observed using a fluorescence microscope (Olympus, Shinjuku, Japan).

### 4.10. Evaluation of Hippocampal Transient Receptor Potential Melastatin 7 Channel Regulation

Transient receptor potential melastatin 7 (TRPM7) channels are expressed at baseline levels under normal conditions in neural cells. However, this channel expression level is increased by external cerebral injury [27]. To compare whether the hippocampal TRPM7 expression level differed between the normal condition and GCI condition, we conducted immunofluorescence for TRPM7 and counter-stained with NeuN. Thirty μm-thick brain slices were incubated with a mixture of goat anti-TRPM7 (diluted 1:50, Abcam, Cambridge, United Kingdom) and mouse anti-NeuN (diluted 1:500, Billerica, Millipore Co., Burlington, MA, United States) for 15~18 h at 4 °C. Between the primary and secondary antibodies reaction, the sections were washed with 0.01 M PBS three times for 10 min. The sections were reacted with a mixture of Alexa Fluor-488 conjugated donkey, anti-mouse IgG and Alexa Fluor-594 conjugated donkey, anti-goat IgG (diluted 1:250, Invitrogen, Carlsbad, CA, United States) for 2 h. The TRPM7 fluorescence signal was quantified by using ImageJ, and used for statistical analysis.

### 4.11. Statistical Analysis

All the results of the present experimental cohorts were shown as the mean value ± standard error of mean (SEM). Comparisons between vehicle-treated and carvacrol-treated rats were conducted using the non-parametric Mann–Whitney U test. Statistical significance was set at *p* < 0.01.

## Figures and Tables

**Figure 1 cells-07-00231-f001:**
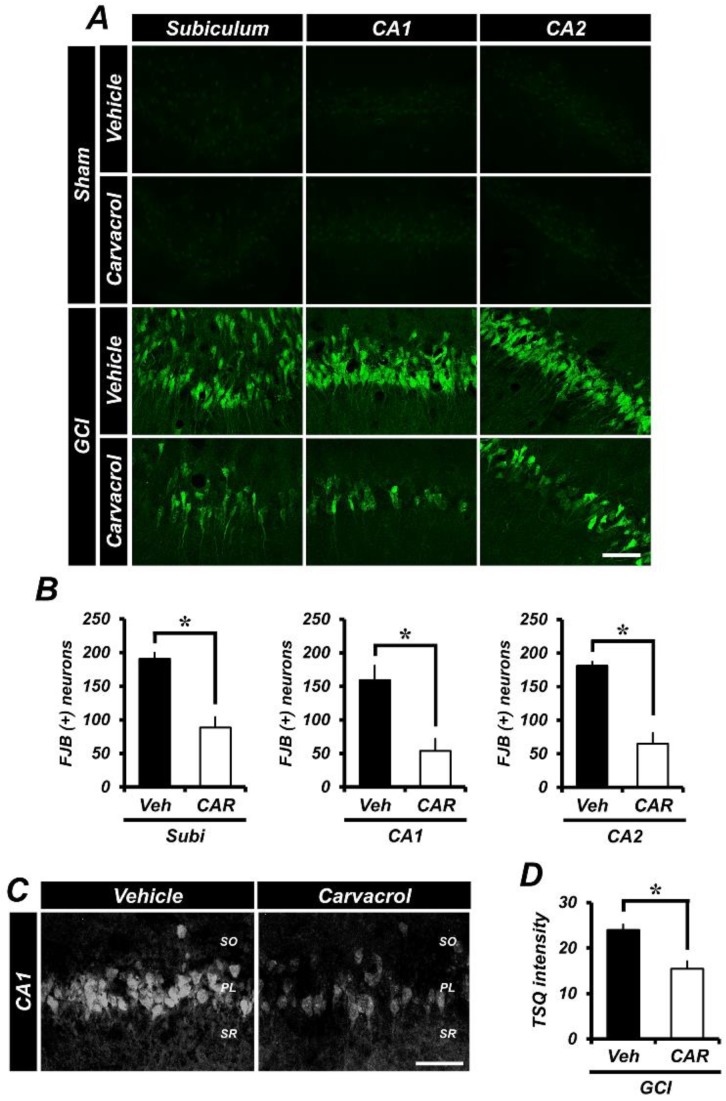
Carvacrol administration decreased degenerating neurons and hippocampal zinc accumulation after global cerebral ischemia (GCI). (**A**) Brain sections were stained with Fluoro-Jade B (FJB, green signal) to detect degenerating neurons in the hippocampal subiculum (Subi), Cornus Ammonis 1 (CA1), and Cornus Ammonis 2 (CA2) after GCI. There were no FJB-positive degenerating hippocampal neurons in the sham-operated groups. FJB-positive neurons were observed in the Subi, CA1, and CA2 of the hippocampus after GCI. Administration of carvacrol after GCI once per day for three days reduced FJB-positive neurons in the Subi, CA1, and CA2 regions. Scale bar = 50 μm. (**B**) The counted FJB positive degenerating neurons. There was a statistically significant difference between the GCI vehicle group and the GCI carvacrol group in the Subi, CA1, and CA2 regions. When compared to the GCI vehicle group, the GCI carvacrol group shows reduced degenerating neurons. (GCI vehicle: *n* = 5; GCI carvacrol: *n* = 5). Data are the mean ± standard error of mean (SEM), * *p* < 0.01. (**C**) Zinc-specific *N*-(6-methoxy-8-quinolyl)-*para*-toluenesulfonamide (TSQ) fluorescence photomicrographs in the hippocampal CA1 region at three days after GCI. The bright white fluorescence signal indicates zinc accumulation. Zinc accumulation was higher in the GCI vehicle group. However, carvacrol treatment reduced zinc accumulation in the hippocampal CA1. Scale bar = 50 μm. (**D**) Quantified TSQ fluorescence intensity signal (GCI vehicle: *n* = 4; GCI carvacrol: *n* = 4). Data are the mean ± SEM, * *p* < 0.01.

**Figure 2 cells-07-00231-f002:**
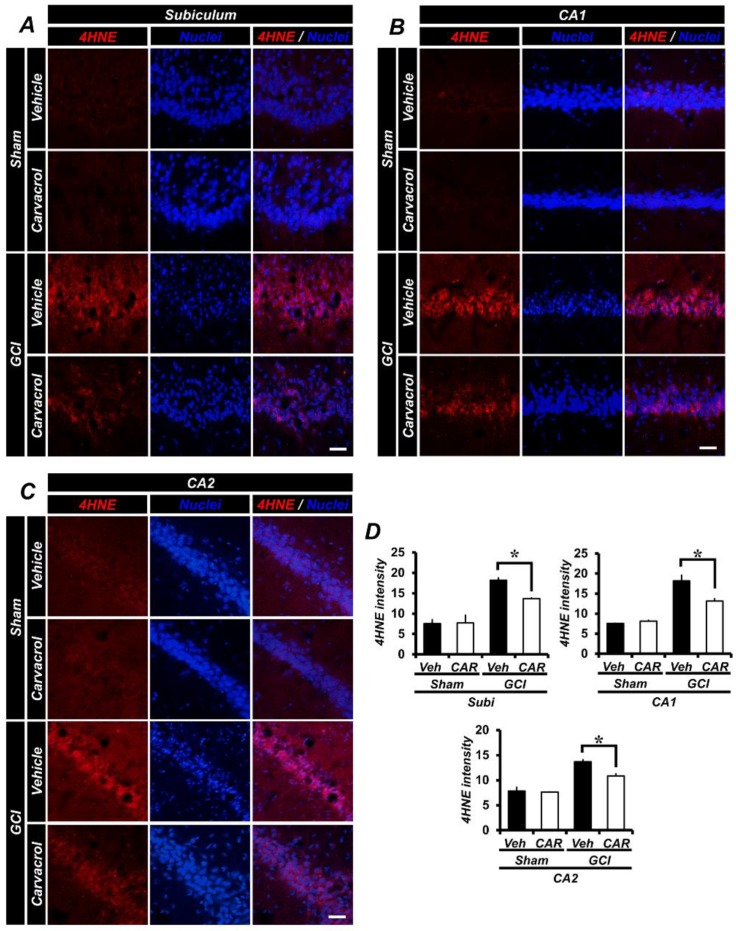
GCI-induced hippocampal oxidative damage was evaluated by 4-hydroxy-2-nonenal (4HNE; red signal) staining, counter-stained with 4’,6-diamidino-2-phenylindole (DAPI; for detecting nuclei; blue signal) at the hippocampal subiculum, CA1, and CA2 regions (sham vehicle: *n* = 5; sham carvacrol: *n* = 4; GCI vehicle: *n* = 5; GCI carvacrol: *n* = 5). (**A**–**C**) The sham-operated groups displayed a minimal 4HNE fluorescence signal in the hippocampal subiculum, CA1, and CA2 regions. The 4HNE fluorescence red signal was outstandingly increased in the GCI vehicle group. This signal was decreased in the GCI carvacrol group at three days after ischemia surgery. Scale bar = 50 μm. (**D**) Degree of quantified neuronal oxidative damage in the hippocampal subiculum, CA1, and CA2 areas. Data are the mean ± SEM, ** p* < 0.01.

**Figure 3 cells-07-00231-f003:**
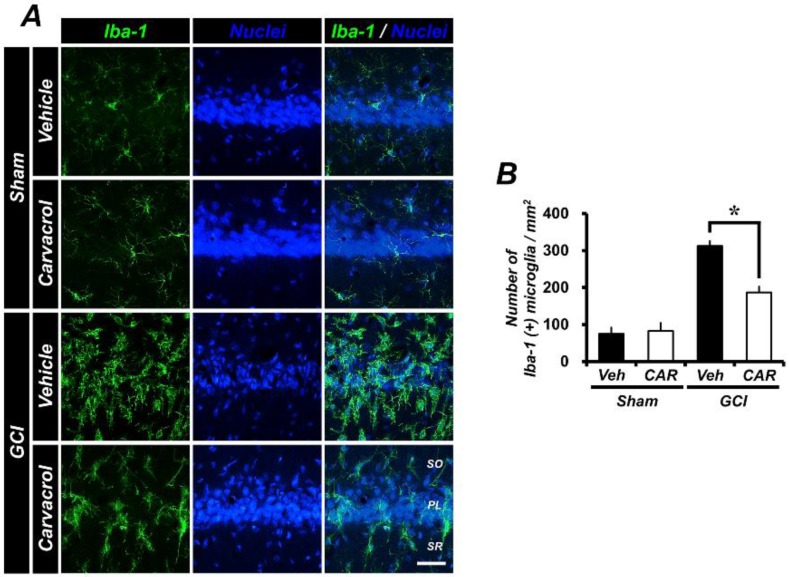
GCI-induced microglial activation by initiating inflammatory cascades was observed using ionized calcium-binding adaptor molecule (Iba1) staining (green signal), counter-stained with DAPI (sham vehicle: *n* = 5; sham carvacrol: *n* = 4; GCI vehicle: *n* = 5; GCI carvacrol: *n* = 5). (**A**) The sham-operated groups showed a microglia fluorescence signal, as well as the number of microglial cells in the hippocampal CA1 region associated with baseline resting conditions. Under the ischemic condition, the number of Iba1-positive microglial cells and the intensity of fluorescence signals were significantly increased. The group that underwent carvacrol administration after GCI showed reduced microglial activation. Scale bar = 50 μm. (**B**) The number of microglial cells in the hippocampal CA1 region. Data are the mean ± SEM, * *p* < 0.01.

**Figure 4 cells-07-00231-f004:**
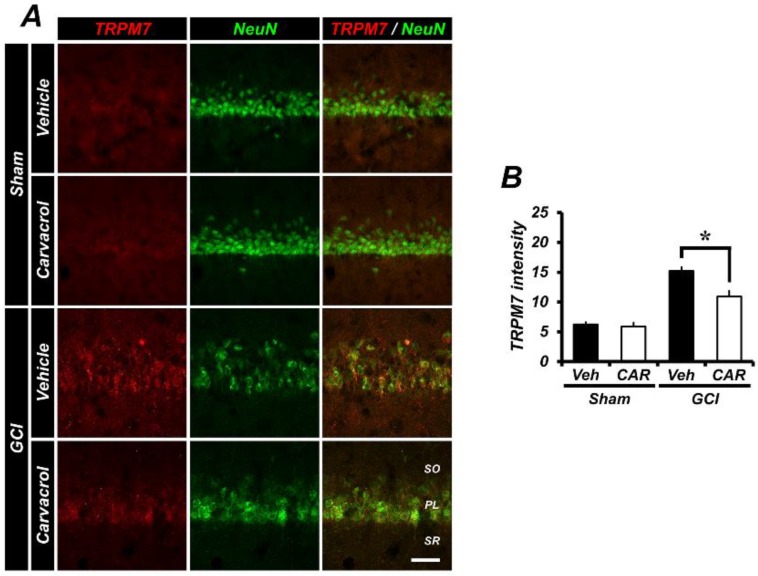
The transient receptor potential melastatin 7 (TRPM7) channel expression level was verified by performing TRPM7 staining and counter-staining with neuronal nuclei. (**A**) In the sham group, the TRPM7 channel expression level showed baseline intensity. This red TRPM7 fluorescence signal was increased in the ischemia vehicle group. Carvacrol administration after GCI for three days showed a neuroprotection effect in the hippocampal CA1 region via inhibition of the TRPM7 channel. Scale bar = 50 μm. (**B**) Quantified TRPM7 channel intensity (red color) in CA1 (sham vehicle: *n* = 5; sham carvacrol: *n* = 4; GCI vehicle: *n* = 5; GCI carvacrol: *n* = 5). Data are the mean ± SEM, * *p* < 0.01.

**Figure 5 cells-07-00231-f005:**
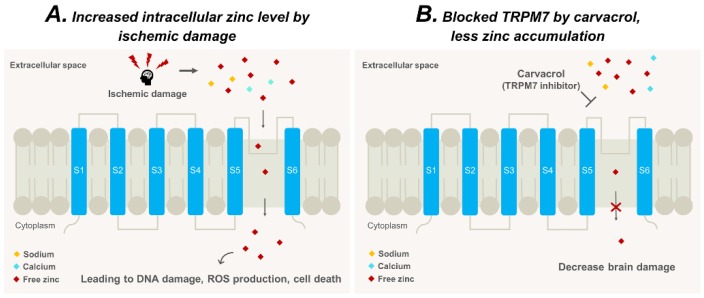
This schematic illustration indicates carvacrol action via inhibition and downregulation of the TRPM7 channel. (**A**) Ischemic brain insult causes an increase in extracellular sodium, calcium, and free zinc levels. These ions could be moved into the intraneuronal space via various proton-sensitive cation channels, such as the acid-sensing ion channel (ASIC), the *N*-methyl-d-aspartate receptor (NMDAR), and transient receptor potential melastatin 7 (TRPM7). In particular, the TRPM7 channel mediates zinc ion regulation. (**B**) Carvacrol administration inhibits intraneuronal zinc accumulation via the TRPM7 channel. Because of this agent’s mechanism, zinc accumulation is decreased by inhibiting TRPM7, resulting in decreased hippocampal neuron death after an ischemic insult.

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
