# Peer review of "Carvacrol Attenuates Hippocampal Neuronal Death after Global Cerebral Ischemia via Inhibition of Transient Receptor Potential Melastatin 7"

_cells, 2018, doi:10.3390/cells7120231_

Round 1

Reviewer 1 Report

The manuscript titled “Carvacrol Attenuates Hippocampal Neuronal Death after Global Cerebral Ischemia via Inhibition of Transient Receptor Potential Melastatin 7” by Hong and colleagues focuses on am important aspect of adult brain injury, which is transient cerebral ischemia brain injury. The manuscript is well written, with mostly clear and well written section.s

Major comments

The n numbers used in this study are very small. Have the authors performed power calculation to demonstrate that these numbers are appropriate for such a study? Furthermore, with such small numbers, it is not appropriate to use a parametric test like ANOVA. Therefore, the authors need to reanalyse the data using non-parametric tests such as Mann-Whitney and Kruskal Wallis.

Minor comments

Results

In the section 2.1. the authors mention that there is no FJB staining present in the sham-operated groups, however, the micrographs of these groups do not appear to be included in the figure 1, please add.

Methods

In the section 4.2. the authors fail to mention whether they used both male and female Sprague-Dawley rats or not.

In the section 4.3. the authors need to describe the transient occlusion methodology more thoroughly. What were the common carotid arteries clamped with? How successful are the authors in avoiding stimulation of the vagus nerve? Did the MAP return to baseline? Furthermore, DMSO is known to be toxic. As DMSO was used for the Carvacrol solution preparation, DMSO should also have been included in the vehicle saline group. Could the authors please clarify why 0.1% DMSO was not included in the vehicle treatment? Did the rats have access to food and water during the recovery period?

Author Response

Reviewer #1: The manuscript titled “Carvacrol Attenuates Hippocampal Neuronal Death after Global Cerebral Ischemia via Inhibition of Transient Receptor Potential Melastatin 7” by Hong and colleagues focuses on am important aspect of adult brain injury, which is transient cerebral ischemia brain injury. The manuscript is well written, with mostly clear and well written section.

Major comments:

The n numbers used in this study are very small. Have the authors performed power calculation to demonstrate that these numbers are appropriate for such a study? Furthermore, with such small numbers, it is not appropriate to use a parametric test like ANOVA. Therefore, the authors need to reanalyze the data using non-parametric tests such as Mann-Whitney and Kruskal Wallis

<Response: We appreciate this reviewer’s comments. As this reviewer suggested, we reanalyzed the data using the non-parametric Mann-Whitney U test. We added this information about the statistical analysis of these data in Materials and Methods of the revised manuscript.>

Minor comments:

In the section 2.1. the authors mention that there is no FJB staining present in the sham-operated groups, however, the micrographs of these groups do not appear to be included in the figure 1, please add.

<Response: We appreciate this reviewer’s comment and agree with this reviewer’s point. Thus, we added the micrograph (FJB staining) of sham-operated groups in the revised manuscript.>

In the section 4.2. the authors fail to mention whether they used both male and female Sprague-Dawley rats or not.

<Response: We appreciate this comment. In this study, we used Sprague-Dawley male rats. So, we added this information in the revised manuscript.>

In the section 4.3. the authors need to describe the transient occlusion methodology more thoroughly. What were the common carotid arteries clamped with? How successful are the authors in avoiding stimulation of the vagus nerve? Did the MAP return to baseline? Furthermore, DMSO is known to be toxic. As DMSO was used for the Carvacrol solution preparation, DMSO should also have been included in the vehicle saline group. Could the authors please clarify why 0.1% DMSO was not included in the vehicle treatment? Did the rats have access to food and water during the recovery period?

<Response: We appreciate this reviewer’s comments and offer our apologies if the experimental methods were not clear. For the global cerebral ischemia surgery, we clamped the bilateral common carotid arteries using medical clamp (Serrefine, Fine Science Tools co.). To avoid stimulation of the vagus nerve and improve surgical accuracy, we used micro dissecting forceps and surgical microscope (Olympus, SZ61). During all surgical periods, we monitored vitals, MAP and EEG signals. MAP was returned to baseline after unclamping of the common carotid arteries. DMSO (pure concentration) is toxic but diluted with normal saline (to 0.1%) is non-toxic [8]. The concentration of DMSO used in this study was sub-toxic. Control rats for these studies received equal volume of vehicle (0.1% DMSO diluted with normal saline) alone. The manuscript has been modified to prevent any misunderstanding. In addition, all experimental animals undergoing ischemia surgery had free access to food and water ad libitum. We added this information in the revised manuscript.> 

<Reference>

  8.   Brayton, C.F. Dimethyl sulfoxide (DMSO): a review. Cornell Vet 1986, 76, 61-90.

Reviewer 2 Report

This study investigates the effect of carvacrol on hippocampal neurons following global cerebral ischemia. The authors show a decrease in degenerating neurons, microglial activation and zinc translocation which they suggest is due to downregulation of the TRPM7 channel. This is an interesting study and the manuscript is nicely composed.

I have a few minor comments that may improve the manuscript:

Figure 1 A,B – The sham group is not shown for comparison of FJB staining. These should be included since the lack of fluorescence in this group is discussed in the text.

Figure 1 B – For clarity the CGI label could be taken out and the subi, CA1, CA2 labels could be shown below the graph.

Figure 2 – 4HNE staining looks similar for GCI vehicle and drug, but there seems to be a reduction in DAPI – how can this be? The reduction of 4HNE in the bar graphs is not obvious in the images.  It would be easier to see the effects if the images were split as they are in Figure 3.

Figure 2B - Labels above only two of four bars suggest that the sham data isn’t from the same brain area?

Lines 204-205 – To support the statement that carvacrol significantly reduces channel expression, RT-PCR for TRPM7 should be performed in these brain areas.

Figure 4 A – Vehicle images look like they’re taken from a different brain region as there is a thicker band of NeuN.

Author Response

Reviewer #2: This study investigates the effect of carvacrol on hippocampal neurons following global cerebral ischemia. The authors show a decrease in degenerating neurons, microglial activation and zinc translocation which they suggest is due to downregulation of the TRPM7 channel. This is an interesting study and the manuscript is nicely composed.

I have a few minor comments that may improve the manuscript:

Figure 1 A, B – The sham group is not shown for comparison of FJB staining. These should be included since the lack of fluorescence in this group is discussed in the text

<Response: We appreciate this reviewer’s comments. We added the micrograph (FJB staining) of sham-operated groups in the revised manuscript.>

Figure 1 B – For clarity the CGI label could be taken out and the subi, CA1, CA2 labels could be shown below the graph.

<Response: We appreciate this comment. We corrected it.>

Figure 2 – 4HNE staining looks similar for GCI vehicle and drug, but there seems to be a reduction in DAPI – how can this be? The reduction of 4HNE in the bar graphs is not obvious in the images. It would be easier to see the effects if the images were split as they are in Figure 3.

<Response: We appreciate this reviewer’s comments. Figure 2 was spilt to 4HNE, nuclei (DAPI) and merged image within each hippocampal region. We corrected it in the revised manuscript.>

Figure 2B - Labels above only two of four bars suggest that the sham data isn’t from the same brain area?

<Response: We appreciate this comment. 4HNE staining data as well as all histological data in Sham and GCI groups were evaluated in hippocampal same regions. We apologize for that misunderstanding, and have modified labels position.>

Lines 204-205 – To support the statement that carvacrol significantly reduces channel expression, RT-PCR for TRPM7 should be performed in these brain areas.

<Response: We appreciate this reviewer's suggestion. However, we are not ready this for this experiment in the current study but plan to address it in a future study.>

Figure 4 A – Vehicle images look like they’re taken from a different brain region as there is a thicker band of NeuN.

<Response: We appreciate this comment. There are slight differences between experimental animals, but region of Figure 4A is the same. We modified vehicle images for clarity.>

Reviewer 3 Report

The paper by Hong et al. entitled "Carvacrol Attenuates Hippocampal Neuronal Death after Global Cerebral Ischemia via Inhibition of Transient Receptor Potential Melastatin 7" shows that Carvacrol palys a neuroprotective and anti-inflammatory role against ischemia via downregulation of TRPM7. This is a nicely executed study that is easy to follow. A few minor comments follow below:

1. In Figure 3, the authors instead of showing "microglia intensity" should show "number of microglial cells/mm2"

2. in Figure legend 5, (B) is not very clear; what do the authors mean by TRPM7 channel activation inhibits carvacrol administration?" Please rephrase, this sentence does not make sense. 

Author Response

Reviewer #3: The paper by Hong et al. entitled "Carvacrol Attenuates Hippocampal Neuronal Death after Global Cerebral Ischemia via Inhibition of Transient Receptor Potential Melastatin 7" shows that Carvacrol plays a neuroprotective and anti-inflammatory role against ischemia via downregulation of TRPM7. This is a nicely executed study that is easy to follow. A few minor comments follow below:

In Figure 3, the authors instead of showing "microglia intensity" should show "number of microglial cells/mm2"

<Response: We appreciate this reviewer’s comment. We reanalyzed the assessment of microglial activation and added it to the revised manuscript.>

In Figure legend 5, (B) is not very clear; what do the authors mean by TRPM7 channel activation inhibits carvacrol administration?" Please rephrase, this sentence does not make sense.

<Response: We appreciate this comment and agree with this reviewer’s point. We deleted and re-wrote this sentence as follow: “Carvacrol administration inhibits intraneuronal zinc accumulation via the TRPM7 channel”.>

Round 2

Reviewer 1 Report

The authors have addressed all the comments appropriately.

Reviewer 2 Report

I am happy with the changes that the authors have made.